# Induction of Collagenolytic MMP-8 and -9 Tissue Destruction Cascade in Mouth by Head and Neck Cancer Radiotherapy: A Cohort Study

**DOI:** 10.3390/biomedicines12010027

**Published:** 2023-12-21

**Authors:** Ella Brandt, Mutlu Keskin, Ismo T. Räisänen, Taina Tervahartiala, Antti Mäkitie, İlknur Harmankaya, Didem Karaçetin, Jaana Hagström, Jaana Rautava, Timo Sorsa

**Affiliations:** 1Department of Oral and Maxillofacial Diseases, University of Helsinki and Helsinki University Hospital, 00290 Helsinki, Finlandtimo.sorsa@helsinki.fi (T.S.); 2Oral and Dental Health Department, Altınbaş University, 34147 Istanbul, Turkey; 3Department of Otorhinolaryngology—Head and Neck Surgery, University of Helsinki and Helsinki University Hospital, P.O. Box 263, 00029 Helsinki, Finland; antti.makitie@helsinki.fi; 4Research Program in Systems Oncology, Faculty of Medicine, University of Helsinki, 00014 Helsinki, Finland; 5Department of Radiation Oncology, Başakşehir Çam and Sakura City Hospital, 34480 Istanbul, Turkey; 6Department of Pathology, Helsinki University Hospital, 00014 Helsinki, Finland; 7Department of Oral Pathology and Radiology, University of Turku, 20520 Turku, Finland; 8Department of Medicine and Dental Medicine, Karolinska Institute, 171 77 Stockholm, Sweden

**Keywords:** head and neck cancer, radiotherapy, biomarker, matrix metalloproteinase, aMMP-8, periodontitis

## Abstract

The effect of head and neck cancer (HNC) radiotherapy (RT) on biomarkers is not known but there is a lot of potential for gaining more precise cancer treatments and less side effects. This cohort study investigated the levels and molecular forms of the matrix metalloproteinase (MMP) -8 and -9, tissue inhibitor of metalloproteinase (TIMP)-1, myeloperoxidase (MPO) and interleukin (IL)-6 in mouth-rinse samples as well as the clinical periodontal status in HNC patients (*n* = 21) receiving RT. Complete periodontal examinations were performed pre-RT and one month after RT. Mouth-rinse samples (pre-RT, after six weeks of RT and one month after RT) were assayed using a point-of-care-kit (PerioSafe^®^/ORALyzer^®^ (Dentognostics GmbH, Jena, Germany)) for active MMP-8 and ELISA analysis for total MMP-8 and -9, MPO, TIMP-1, and IL-6 levels. Molecular forms of MMP-9 were assessed by gelatinolytic zymography and MMP-8 by western immunoblot. Significant changes were observed between the three time points in the mean levels of active and total MMP-8, active MMP-9, and IL-6. Their levels increased during the RT and decreased after the RT period. The aMMP-8 levels stayed elevated even one month after RT compared to the pre-RT. Clinical attachment loss, probing depths, and bleeding on probing were increased between pre- and post-calculations in periodontal status. Elevated inflammatory biomarker levels together with clinical recordings strongly suggest that RT eventually increases the risk to the periodontal tissue destruction by inducing the active proteolytical MMP-cascade, and especially by prolonged activity of collagenolytic aMMP-8. Eventually, the aMMP-8 point-of-care mouth-rinse test could be an easy, early detection tool for estimating the risk for periodontal damage by the destructive MMP-cascade in HNC patients with RT treatment.

## 1. Introduction

Radiotherapy (RT) is commonly used in head and neck cancer (HNC) treatment. It is used either alone or in combined treatment strategies involving surgery with adjuvant radiotherapy, and radio-chemotherapy [1]. However, radiation to the head and neck area has a high rate of diverse acute and late side effects that predispose patients to infections and treatment delays as well as a decreased quality of life [2]. RT causes permanent damage to the jawbone and microvascular system, causing an impaired wound-healing capacity and tissue regeneration. Oral health is also challenged by the changes in defensive immune responses, reduced saliva production, challenges in basic functions and oral hygiene, and altered oral microbiota [3]. The risk for tooth loss and clinical attachment loss have been shown to increase in the high-dose radiated sites of HNC patients [4,5].

Periodontal diseases, which include gingivitis and periodontitis, are typically slowly progressing plaque-induced polymicrobial diseases leading to inflammation and destruction of the gum tissues, periodontal ligament, and alveolar bone [6]. Periodontal diseases develop from the interactions between local microbiota and the host-immune response, with multiple contributing etiological risk factors. Poor oral hygiene and tobacco usage are the main risk factors for periodontal diseases. Studies have indicated a positive correlation between periodontitis and several cancers, especially oral/esophageal cancers [7,8,9].

The effect of RT on biomarkers, genomics, and proteome profiles are not well known but nowadays increasingly studied. Oncological biomarkers show increasing amounts of clinical applications as they can predict treatment outcomes, monitor the response to therapy, or treatment side effects [10]. Potential salivary biomarkers have been identified but much more knowledge is needed to gain validation and reliability for their clinical applications [11]. Saliva-based liquid biopsies and mouth-rinse sampling are promising approaches for easy, non-invasive, and pain-free sampling, early diagnosis, and the monitoring of disease progression and treatment response in HNC patients [12].

Matrix metalloproteinases (MMPs) are a family of host-derived, genetically distinct but structurally related endopeptidases including, i.e., collagenases, gelatinases, stromelysins, and matrilysins, and they have a critical role in many physiological and inflammatory processes. MMPs related to the periodontal diseases are produced most importantly by neutrophils and their primary function is tissue remodeling by degrading various proteins in the extracellular matrix [13]. Latent proMMPs are enzymatically inactive. Transforming into active, catalytically competent active MMPs (aMMPs) requires molecular cleavage by independent or co-operative cascades involving oxygen-derived free radicals, pathogen and host proteases, as well as inflammatory mediators [14].

The regeneration of connective tissues includes balance between MMPs and their inhibitors, known as tissue inhibitors of metalloproteinases (TIMPs). In inflammation, this balance can be disrupted, leading to excessive MMP activity and often irreversible tissue destruction and degeneration. Breaking down collagen fibers enables more inflammatory cells to migrate to the infection site but with hindsight disrupts the integrity and strength of the connective tissue.

MMP-8 and -9 are the most abundant MMPs in periodontal tissues and oral fluids and when activated, they are responsible for the breakdown and remodeling of the extracellular matrix, such as collagen, elastin, and basement membranes. aMMP-8 (i.e., collagenase-2/neutrophil collagenase) is a catalytically competent collagenolytic protease and a major mediator of irreversible tissue destruction in periodontitis and can be utilized as a biomarker of periodontitis [15,16]. An increase in the expression of aMMP-8 in oral fluid diagnostic biomarker analysis shows the onset of the periodontal diseases and is related to the active progression of the disease. The new 2017 classification systems’ stages and grades of periodontitis and peri-implantitis assessment highlights the need for validated biomarkers for early detection of periodontitis [17]. Diagnostical attempts to use total MMP-8 to discriminate healthy and diseased periodontium have shown contrasting results [18,19,20,21].

Cytokines, like IL-6, are modulators of both homeostasis and inflammation. Pathogens and barrier site stimuli stimulate cytokines to start proinflammatory events [22]. Myeloperoxidase (MPO) is an enzyme primarily found in neutrophils. MPO takes part in the immune defense reaction against microbial pathogens with its powerful pro-oxidative properties but also by independently modulating inflammatory responses, such as oxidatively activating and inactivating other enzymes and their inhibitors, such as MMPs and TIMPs [23].

The objective of this study was to investigate the effect of HNC RT on the levels and molecular forms of MMP-8 and -9, TIMP-1, MPO, and IL-6 in mouth-rinse samples, as well as clinical periodontal status before and one month after the RT. We hypothesize that these inflammatory biomarkers and aMMP-8 point-of-care test react to inflammation caused by RT and correlate with weakened periodontal health.

## 2. Materials and Methods

### 2.1. Patient Selection and Study Design

Twenty-one HNC patients out of 24 recruited patients gave all three time-point (pre-RT, after six weeks of RT and one month after RT) mouth-rinse samples and were included in the study. All patients were treated in Başakşehir Çam and Sakura City Hospital, Istanbul, Turkey. Inclusion criteria were histologically confirmed as head and neck carcinoma; at least 21 years of age; presence of at least five teeth; and RT planned for HNC treatment. Exclusion criteria were the inability to give all three mouth-rinse samples; cancer and/or radiotherapy history, immune-associated disorders (i.e., chronic inflammatory diseases such as lupus erythematosus, rheumatoid arthritis, multiple sclerosis, Crohn’s disease, and HIV); patients who received bisphosphonate therapy; and patients whose RT process was interrupted.

Patients’ demographic data and their medical history including the systemic diseases, medications, and smoking habits of the patients were recorded. Patients’ treatment plans were decided individually for each patient by expert medical oncologists following the National Comprehensive Cancer Network^®^ (NCCN^®^) guidelines considering their systemic conditions, pathology of the HNC, and imaging reports of their cancer [24]. Periodontal health was examined by an experienced periodontist before and one month after RT. Patients were instructed to maintain good mouth hygiene during HNC treatment. The reporting of this study conforms to STROBE guidelines [25].

### 2.2. The Collection of Oral Rinse and aMMP-8 Analysis

HNC patients had their mouth-rinse samples collected before RT (pre-RT), at the end of the six-week-RT (post-RT), and one month after RT with the non-invasive, clinically available aMMP-8 chair-side mouth-rinse test (PerioSafe^®^, Dentognostics GmbH, Jena, Germany). The test sensitivity was in the range of 75–85% and specificity 80–90% with a 20 ng/mL cut-off [26].

Patients were instructed to restrain eating or brushing teeth one hour before mouth-rinse sampling. First, the patients rinsed their mouth with drinking water for 30 s and then spit the water out. After 60 s pause, they rinsed again with 5 mL of distilled water for 30 s and then spit it into a measuring cap. Following the manufacturer’s instructions, the sample was taken from the measuring cup with a syringe and filtered and transferred to the aMMP-8 Periosafe^®^ kit test cassette. The quantitative aMMP-8 result was analyzed within 5 min by the digital reader ORALyzer^®^ (Dentognostics GmbH, Jena, Germany). The left-over mouth-rinse was transferred info Eppendorf tubes and stored at −70 °C for laboratory analysis.

### 2.3. ELISA Analysis, Western Immunoblot and Gelatine-Substrate Zymography

The concentrations of MMP-8 and -9, TIMP-1, and MPO, as well as IL-6, were determined by a commercially available enzyme-linked immunosorbent (ELISA) kit specific for each biomarker according to the protocol of the manufacturer (Human IL-6 Quantikine^®^ ELISA kit, R&D Systems, Minneapolis, MN, USA). The minimum detectable concentrations were 0.013 ng/mL for total MMP-8, 0.156 ng/mL for total MMP-9, 1.25 ng/mL for TIMP-1, 0.324 ng/mL for MPO, and 0.70 pg/mL for IL-6.

The molecular forms of MMP-8 were detected by a modified enhanced chemiluminescence (ECL) Western blotting kit according to protocol recommended by the manufacturer (Amersham ECL Western Blotting Detection Kit, catalogue number RPN2108, Cytiva, Marlborough, MA USA). Samples were run on 7–10% SDS-polyacrylamide gels and transferred to nitrocellulose membranes (Schleicher & Schüll, Dassel, Germany). Nonspecific binding was blocked with 5% skim milk for 90 min at 37 °C. The membranes were incubated with a rabbit polyclonal antibody (1:1000 dilution) against MMP-8 for 3 h at 37 °C and followed by peroxidase-conjugated goat anti-rabbit immunoglobulins (1:200 dilution; DAKO A/S, Glostrup, Denmark) for 1 h at 22 °C. After washing, the blot was developed with a solution of 60 mg/mL diaminodenzidine tetrahydrochloride in 50 mM Tris-HCl, pH 8.0, and 0.003% H_2_O_2_ [26,27].

The immunoblots were quantified by a Bio-Rad Model GS-700 Imaging Densitometer using Bio-Rad Quantity One program (Bio-Rad Laboratories Inc., Hercules, CA, USA).

Gelatin-substrate zymography was used to detect and quantify the presence of the gelatinases, particularly MMP-9, in mouth-rinse samples, using technique as previously described [27].

### 2.4. Statistics

The periodontal parameters clinical attachment loss (CAL), probing depth (PD), bleeding on probing (BOP %), visual plaque index (VPI), furcation involvement, and mobility were examined before RT and one month after RT. The significances between the periodontal measurements were calculated by the paired *t*-test with Cohen’s d for effect size. In addition, biomarker levels of aMMP-8, total MMP-8, aMMP-9, pMMP-9, total MMP-9, MMP-9 fragments, IL-6, TIMP-1, and MPO were measured before RT, after six weeks of RT, and one-month after RT. The significant differences in the mean levels of the biomarkers between the three time points were tested by repeated-measures ANOVA analysis with partial Eta squared for effect size, and following pairwise comparisons were adjusted for multiple comparisons by the Bonferroni post hoc test. A two-tailed *p*-value below 0.05 was considered statistically significant. Statistical analyses were made using the SPSS version 29.0 (IBM SPSS Statistics for Windows, IBM Corp., Armonk, NY, USA).

## 3. Results

All patients had HNC, and their treatment included RT with curative intent. Table 1 and Table 2 present the characteristics of the patients including demographic and oncological information. The deteriorating impact of RT on the periodontium is shown in Table 3.

The study comprised 16 (76%) males and 5 (24%) females. All patients had a history of smoking (≥10 cigarettes per day) of more than 5 years but had been asked to quit smoking before cancer treatments. Patients’ mean age was 55 years (range 28–84) at the time of HNC diagnosis. The location of the HNCs varied and the three most common locations were the nasopharynx (29%), larynx (19%), and oropharynx (14%).

RT was administered for six weeks for each patient according to standard treatment guidelines using the IMRT technique with 6 MV photon energy and the Elekta Synergy Linear Accelerator (Elekta Oncology^®^, Crawley, UK) as previously described [26]. Patients received a total RT dose of 65 Gy (range 50–70 Gy). Fifty-two percent of the patients received post-operative RT (average total dose 64 Gy) and 48% were treated with primary RT (average total dose 66 Gy). Surgery aimed at radical tumor resection in each case. A total of eleven patients (52%) received additional chemotherapy. Concomitant chemotherapy was administered using Cisplatin, and neo-adjuvant chemotherapy for nasopharyngeal carcinoma patients using Gemcitabine and Cisplatin.

There was a significant increase in CAL (mm), PD (mm), and BOP (%) (Table 3). Furthermore, the change in CAL between post- and pre-RT measurements (the site of greatest loss measured) ranged from 1 to 3 mm and was on average 1.4 mm, with standard deviation of 0.6. That corresponds to grade C (=rapid progression) according to the classification system of periodontitis [17]. The change in VPI, the number of teeth with mobility, and the number of teeth with furcation were not significant.

Significant changes were observed between the three time points (before RT (pre-RT), after six weeks of RT (post-RT), and one-month after RT) in the mean levels of aMMP-8, total MMP-8, aMMP-9, and IL-6 (Figure 1, Figure 2 and Figure 3). The mean levels of these biomarkers increased during the RT and decreased after the RT period. Pairwise testing revealed a significant increase in the mean levels of aMMP-8 between the pre- and post-RT measurements, but the decrease was not significant at one-month after the RT. Furthermore, the increase during RT and the decrease after RT were not significant for total MMP-8. On the other hand, the pairwise testing showed that the decrease one-month after RT in the mean levels of aMMP-9 and IL-6 was significant, but the increase during the RT was not. Moreover, the mean levels of proMMP-9, total MMP-9, MMP-9 fragments, and MPO were increasing during the RT and decreasing after it had finished—but the differences in the mean levels were not significant between the three time points. The mean level of TIMP-1 decreased during and after the RT—but again the difference was not significant between the three time points. Figure 4 shows the gelatinolytic activity in zymography between three studied time-points in one patient. MMP-9 activation and fragmentation to lower molecular size species during the RT can clearly be detected in the zymography. Western blot analysis revealed the presence of *Treponema denticola* chymotrypsin-like protease dentisilin and *Porphyromonas gingivalis*) gingipain at steady levels in mouth-rinse samples.

## 4. Discussion

The present study was conducted as a prospective cohort study to assess the effect of HNC RT on mouth-rinse biomarkers MMP-8, MMP-9, TIMP-1, MPO, and IL-6 and clinical periodontal status. The results showed significant changes between the three time points (pre-RT, after six weeks of radiotherapy, and one month after radiotherapy) in the mean levels of both active and total MMP-8, active MMP-9, and IL-6. This indicates an increased risk for periodontitis by the upregulation of the active collagenolytic MMP-dependent tissue destruction cascade due to the HNC radiotherapy. Additionally, periodontal health worsened rapidly as there were increases in both CAL and PD one month after RT.

Our findings support and further extend the earlier observations showing an increase in aMMP-8 levels and clinical attachment loss due to the RT in HNC patients [28]. In accordance with the present results, MMP-8 have been demonstrated to be upregulated in whole-saliva samples in radiated HNC patients compared to healthy controls [29]. Active MMP-8 levels and the neutrophil/lymphocyte ratio have been shown to increase in the first three weeks of HNC RT, suggesting inflammation both locally and systemically [26]. The present study confirmed and thus further extended these observations showing how the aMMP-8 levels continued increasing with the RT and decreased when RT ended. Additionally, this study revealed that also MMP-9 is activated and fragmented by RT-induced upregulation. This RT-promoted MMP-8 and -9 upregulation is eventually mediated, at least in part, by the co-operative actions of IL-6, MPO, and microbial factors.

Typical periodontitis is a chronic disease that develops slowly. According to the periodontal staging and grading classification [17], the evidence of disease progression (radiographic bone loss or CAL) of ≥2 mm over 5 years is considered as rapid progression. The fact that the patients in this study experienced CAL (at the site of greatest loss) of on average 1.4 mm (SD 0.6) in between, before and one month after RT reflects the rapid and/or highly accelerated periodontal breakdown. This is in line with previous studies that showed clinical attachment loss occurring due to the RT [5,28,30].

MMP-8 is the most prevalent collagenolytic protease in the diseased periodontium and aMMP-8 the major mediator of periodontitis [15]. As the aMMP-8 levels stayed elevated after RT, it is possible that collagenolytic activity and the burden of aMMP-8 predisposes tissue for further periodontal damage. With periodontal disease being increasingly recognized as a contributor in the pathophysiology of several systemic diseases, cancers, as well as risk factor for osteoradionecrosis, it is important to monitor and recognize worsened periodontal status after RT [31,32]. Periodontitis can be detected in real time with chairside/point-of-care aMMP-8 immunotest visually (−/+) and/or quantitatively in 5 min [15].

MMPs are overexpressed in many cancer tissues [33] but the relationship is complex [13]. The MMPs have been mostly considered tumor-promoting because tumor cell tissue invasion and extravasation, angiogenesis and metastasis depend largely on the overexpression of multiple MMPs and have a major impact on the tumor microenvironment [34,35]. In HNCs, MMPs, especially MMP-9, are associated with the degradation of the extracellular matrix, which allows cancer to spread via blood vessels [36]. Further, studies have shown that both MMP-8 and -9 have both tumor-promoting and -suppressing effects in cancers as well [37,38]. It would be beneficial to study if elevated levels of active MMP-8 and -9 in the mouth have tumor-promoting or -suppressing effects in HNC patients that would correlate with treatment success.

Radiotherapy and chemoradiotherapy are known to increase a number of salivary cytokines, i.e., IL-6 and TNF-α and positively correlate with the irradiation dose [39]. Our results similarly demonstrated the increased levels of IL-6 due to RT. Salivary cytokines have been represented as potential diagnostic cancer biomarkers because their significantly elevated levels are found in oral cancer patients, and also in patients with oral potentially malignant disorders, including leukoplakia [40]. Additionally, cytokines have been suggested as potential biomarkers for the early identification of severe chemoradiotherapy-induced oral mucositis, although some studies have shown controversial results [41,42].

MPO and TIMP-1 have been previously demonstrated to be downregulated proteins in tear fluid from radiated HNC patients compared to controls [29]. Our results showed the down-trending curve of TIMP-1 levels but an opposite trend for MPO. It is known that MPO can oxidatively activate proMMP-8 and -9 and inactivate TIMP-1 [43,44]. Mention should be made that the differences in TIMP-1 and MPO levels were not statistically significant in the present cohort. In general, tissue inhibitors of MMPs act primarily through inhibition of MMPs in tumors. However, TIMPs have also independent biological activities such as regulating inflammation and growth in the tumor microenvironment [45].

Periodontopathogens *Treponema denticola* and *Porphyromonas gingivalis* are frequently found in periodontitis patients [46]. All patients in the present study had periodontitis before start of RT. Therefore, it is not surprising that we were able to identify *Treponema denticola* chymotrypsin-like protease dentisilin and *Porphyromonas gingivalis* protease gingipain at steady levels in the mouth-rinse samples. Dysbiotic potent periodontopathogens, such as *Treponema denticola* and *Porphyromonas gingivalis*, can act by secreting proteases that promote host-proMMP-activation [47] and thus may affect the oral and periodontal status in HNC RT patients.

The findings of this study must be considered with some limitations. The study group was quite small and there were variations in HNC locations and treatments. Especially, concomitant chemotherapy might impact the biomarkers independently and therefore affect the results. The lack of a control group of HNC patients not receiving RT is another limitation. Interestingly, the results of this study suggest that even though the carcinoma site and therefore, the primary radiation field is not directly addressed to oral cavity or jaws, patients still experience periodontal damage and effects on oral rinse biomarkers. This can be concluded by the fact that for the majority of the patients’ the tumor location was outside the oral cavity.

Different medications, comorbidities and variations in oral hygiene might have impact on oral fluid biomarker levels. Periodontitis is typically a chronic disease that develops gradually. Patients were instructed to maintain good mouth hygiene and stop smoking during cancer treatment. We can conclude that possible weakened oral hygiene during cancer treatment does not cause the progression of periodontitis or biomarker changes substantially in this study. To lower the impact of additional inflammatory conditions on our results, we excluded all patients with immune-associated diseases from the study. Patients with type 2 diabetes and COPD were included as they had a suitable medication for the condition.

It should be mentioned that chemo-radiotherapy-induced oral mucositis and underwent surgery most probably have some impact to the biomarker levels. Mucositis is an inflammatory response of epithelial mucosa to chemo-radiotherapy-related cytotoxic effects, and it is a common side effect in HNC treatment [42]. Mucositis develops in the radiated area and its severity is directly related to the dose and number of cycles [3]. For all patients who underwent surgery, the radiotherapy was given post-operatively. Therefore, performed surgery might have an effect on inflammation markers during radiotherapy. In future, it is important to investigate the role of primary surgery, including the used reconstruction tissue transfers and concomitant chemotherapy, and their effect on the oral fluid biomarkers as well as periodontal status. More research with larger patient groups is needed to study these phenomena.

We hypothesize that HNC RT can upregulate and activate the collagenolytic MMP-8 and -9 tissue destruction cascade in mouth, utilizing pro-inflammatory cytokines (IL-6), oxidative stress induced by, i.e., MPO-reduced MMPs inhibitory mechanisms, such as the downregulation of their endogenous inhibitors TIMPs, and by potent dysbiotic periodontopathogens’ hydrolytic proteases (dentilisin and gingipain). Importantly, this collagenolytic tissue destruction cascade can be detected in real time with chairside/point-of -care aMMP-8 immunotest visually (−/+) and/or quantitatively in 5 min. The aMMP-8 point-of-care test could be used as a clinical tool outside dentist office for identifying and alarming the periodontal disease and monitoring periodontal health development during and after cancer treatment. Detecting susceptibility to periodontitis early helps both patient and health care professionals to react, before already-experienced tissue destruction, also working as a motivating tool for patients to maintain good mouth hygiene.

## Figures and Tables

**Figure 1 biomedicines-12-00027-f001:**
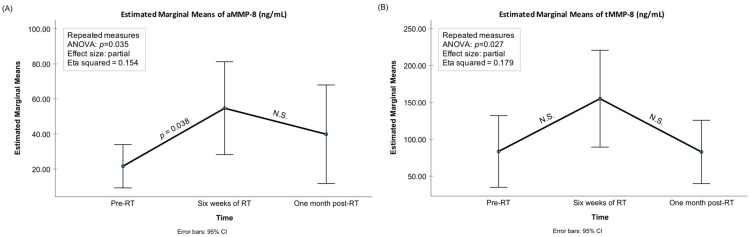
Mean levels of (**A**) active (a)MMP-8 POCT (ng/mL), (**B**) total (t)MMP-8 (ng/mL) with 95% confidence interval bars for the time-points of pre-radiotherapy (RT), after six weeks of RT, and one-month after RT. *p*-values from the repeated-measures ANOVA analysis as well as significant pairwise comparisons adjusted for multiple comparisons (the Bonferroni post hoc test) are presented in the figures. N.S.: Non-significant, i.e., *p* > 0.05.

**Figure 2 biomedicines-12-00027-f002:**
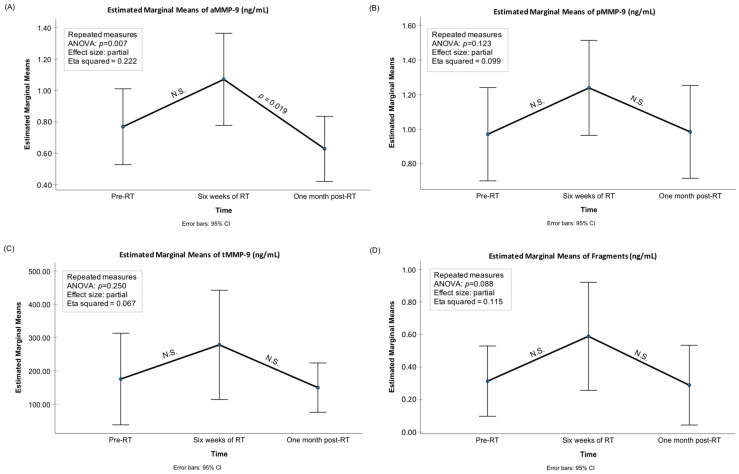
Mean levels of (**A**) active (a)MMP-9 (ng/mL), (**B**) pro (p)MMP-9 (ng/mL), (**C**) total (t)MMP-9 (ng/mL), and (**D**) MMP-9 fragments (ng/mL), with 95% confidence interval bars for the time-points of pre-radiotherapy (RT), after six weeks of RT, and one month after RT. *p*-values from the repeated-measures ANOVA analysis as well as significant pairwise comparisons adjusted for multiple comparisons (the Bonferroni post hoc test) are presented in the figures. N.S.: Non-significant, i.e., *p* > 0.05.

**Figure 3 biomedicines-12-00027-f003:**
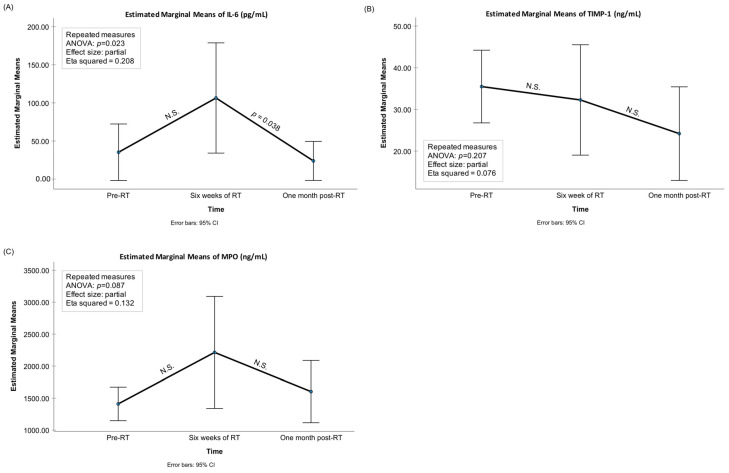
Mean levels of (**A**) IL-6 (pg/mL), (**B**) TIMP-1 (ng/mL), (**C**) MPO (ng/mL) with 95% confidence interval bars for the time-points of pre-radiotherapy (RT), after six weeks of RT, and one-month after RT. *p*-values from the repeated-measures ANOVA analysis as well as significant pairwise comparisons adjusted for multiple comparisons (the Bonferroni post hoc test) are presented in the figures. N.S.: non-significant, i.e., *p* > 0.05.

**Figure 4 biomedicines-12-00027-f004:**
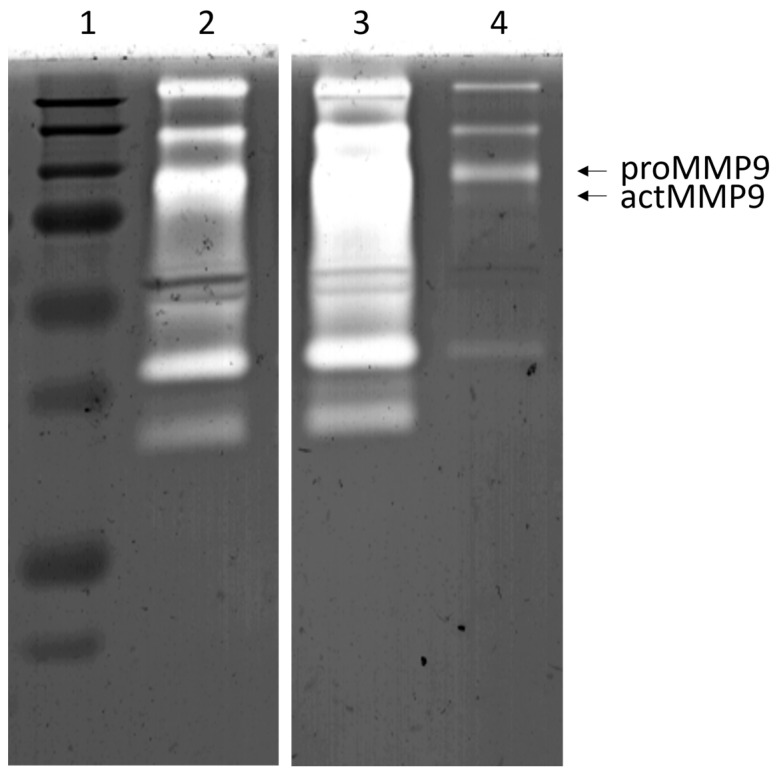
Gelatin-substrate zymographia of gelatinases, representing gelatinolytic activity during RT in one patient. Lane 1: molecular marker; lane 2: before radiotherapy (RT); lane 3: six weeks of RT; lane 4: one-month after RT. Note the increased levels of both pro- and active MMP-9 during RT on the lane 3.

**Table 1 biomedicines-12-00027-t001:** Patient characteristics (*n* = 21).

		*n* (%)
Gender		
	Male	16 (76)
	Female	5 (24)
Comorbidities		
	No additional diseases	9 (43)
	Diabetes type II	5 (24)
	Hypothyroidism	4 (19)
	Chronic obstructive pulmonary disease	3 (14)
	Cardiovascular disease	3 (14)
	Hypertension	1 (5)
Medication		
	Metformin	5 (24)
	Levoythroxine sodium	4 (19)
	Ipratropium bromide	3 (14)
	Acetylsalicylic acid	2 (10)
	Atorvastatin	1 (5)
	Metoprolol	1 (5)
Smoking		
	Yes	21 (100)
	No	0 (0)
Stage of periodontitis [17]		
	Stage I	1 (4)
	Stage II	8 (38)
	Stage III	7 (33)
	Stage IV	5 (24)
Grade of periodontitis [17]		
	Grade A	0 (0)
	Grade B	0 (0)
	Grade C	21 (100)

**Table 2 biomedicines-12-00027-t002:** Oncological characteristics (*n* = 21).

		*n* (%)
Histopathological diagnosis	Squamous cell carcinoma	15 (71)
	Nonkeratinizing undifferentiated nasopharyngeal carcinoma	3 (14)
	Keratinizing nasopharyngeal carcinoma	1 (5)
	Mucoepidermoid carcinoma	1 (5)
	Adenoid cystic carcinoma	1 (5)
Tumor location	Nasopharynx	6 (29)
	Larynx	4 (19)
	Oropharynx	3 (14)
	Hypopharynx	2 (10)
	Parotis	2 (10)
	Mobile tongue	2 (10)
	Another site of oral cavity	1 (5)
	Lip	1 (5)
Treatment Type	Radiotherapy + surgery	8 (38)
	Chemoradiotherapy	8 (38)
	Chemoradiotherapy + surgery	3 (14)
	Radiotherapy	2 (9)

**Table 3 biomedicines-12-00027-t003:** Periodontal parameters prior to before radiotherapy (pre-RT) and one-month after RT. The significance between these two time-points calculated by paired *t*-test for each parameter (*n* = 21).

Variable	Pre-RT (Mean ± SD)	Pre-RT (95% Confidence Interval for Mean)	One Month after RT (Mean ± SD)	One Month after RT (95% Confidence Interval for Mean)	*p*-Value	Effect Size: Cohen’s d (95% Conficence Interval)
Clinical attachment loss (mm) (mean) Probing depth (mm) (mean) Bleeding on probing (%)	2.04 ± 1.25	1.47–2.60	2.16 ± 1.29	1.58–2.75	<0.001	0.96 (−1.47–−0.43)
	3.24 ± 1.04	2.77–3.72	3.33 ± 1.08	2.84–3.82	0.025	−0.53 (−0.98–−0.07)
	51.92 ± 23.79	0.41–0.63	60.84 ± 25.95	0.49–0.73	<0.001	−0.87 (−1.37–−0.36)
Visual plaque index (mean) Mobility (Involved teeth count)	1.24 ± 0.50	1.01–1.47	1.23 ± 0.52	1.00–1.47	0.892	0.03 (−0.40–0.46)
	4.43 ± 5.69	1.84–7.02	4.81 ± 5.93	2.11–7.51	0.134	−0.34 (−0.78–0.10)
Furcation (Involved teeth count)	1.38 ± 2.01	0.47–2.30	1.29 ± 1.82	0.46–2.11	0.329	0.22 (−0.22–0.65)

## Data Availability

Due to the sample size of the study and to conserve participant confidentiality and anonymity, data for this study will not be made available.

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
