# Peer review of "Induction of Collagenolytic MMP-8 and -9 Tissue Destruction Cascade in Mouth by Head and Neck Cancer Radiotherapy: A Cohort Study"

_biomedicines, 2023, doi:10.3390/biomedicines12010027_

Round 1

Reviewer 1 Report

Comments and Suggestions for Authors

Reviewer report

Induction of collagenolytic MMP-8 and - 9 tissue destruction 2 cascade in mouth by head and neck cancer radiotherapy: a cohort study

1. Overview:

The study investigates the impact of head and neck cancer (HNC) radiotherapy on biomarkers, specifically MMP-8 and -9, TIMP-1, MPO, and IL-6, along with clinical periodontal status. The results indicate significant changes in biomarker levels and periodontal health before, during, and after radiotherapy.

2. Suggestions for Improvement:

Statistical Clarity: Provide a clearer presentation of statistical results, including p-values, effect sizes, and confidence intervals in the results and in the Figure legends of Figure 1., Figure 2. And Figure 3.

When you are talking about NCCN and STROBE guidelines, please cite these guidelines.

Please describe what aMMP-8 means the first time you mention it in the text (maybe it is unclear for all of the readers).

Explore and discuss potential confounding factors that might influence the observed changes in biomarkers. Factors such as comorbidities, medications, or variations in oral hygiene practices could be considered and addressed in the analysis.

I detected no plagiarism in this manuscript.

3. Conclusion:

The study offers valuable insights into the impact of HNC radiotherapy on biomarkers and periodontal health. Addressing the suggested improvements will enhance the clarity and impact of the findings.

Reviewer 2 Report

Comments and Suggestions for Authors

The authors have comprehensively investigated the impact of HNC radiotherapy on mouth-rinse biomarkers (MMP-8, MMP-9, TIMP-1, MPO and IL-6) and clinical periodontal status. The reviewer thinks this article is very interesting for Oncology and radiotherapy, however, it needs to be clear and complete in this article. To render the manuscript suitable for publication to Biomedicines, several corrections should be made.

Specific comments.

1. How was the sample size (n=24) determined? 

2. Figures 1-3 are missing, and the quality was not good enough and should be improved.

3. Please provide more information about the western blot analysis.

4. Figure 4, n = 1 or n= 24 ? Please clarify it and add some statistical comparisons. In addition, the quality of the figure needs to be improved.

5. Tables 2-3, Please clarify the relationship between mouth-rinse biomarker and the clinical periodontal status, particularly the impact of types of treatment (such as radiotherapy alone, chemotherapy alone or radiotherapy + chemotherapy + surgery).

Comments on the Quality of English Language

Minor editing of English language required.

Reviewer 3 Report

Comments and Suggestions for Authors

Induction of collagenolytic MMP-8 and - 9 tissue destruction 2 cascade in mouth by head and neck cancer radiotherapy: a cohort study

This is an interesting study confirming earlier observations showing increase in MMP-8 levels and clinical attachment loss due to the radiotherapy in Head & Neck carcinoma patients.

The paper is well written, and structured.  Limitations are indicated (lack of a control group, quite small groups, variations in HNC sites and treatments provided).

I would suggest the authors to specify if statistical analyses for differences among the various groups (for dx, locations, rx+chemotherapy etc)  were done and include the results.

I would also suggest to enlarge the discussion about the clinical utility of such tests for non specialist 

Round 2

Reviewer 2 Report

Comments and Suggestions for Authors

The authors revised the manuscript according to suggestions.